# Long-Term Outpatient Care and Rehospitalizations in Patients after Cardiac Electrotherapy Device Implantation

**DOI:** 10.3390/medicina58020151

**Published:** 2022-01-19

**Authors:** Roman Załuska, Anna Milewska, Anastasius Moumtzoglou, Marcin Grabowski, Wojciech Drygas

**Affiliations:** 1Department of Management and Logistics in Health Care, Medical University of Lodz, 90-131 Lodz, Poland; 2Department of Statistics and Medical Informatics, Medical University of Bialystok, 15-089 Bialystok, Poland; anna.milewska@umb.edu.pl; 3P. & A. Kyriakou Children’s Hospital, 11527 Athens, Greece; anastasius.moumtzoglou@gmail.com; 41st Department of Cardiology, Medical University of Warsaw, 02-097 Warsaw, Poland; marcin.grabowski@wum.edu.pl; 5Department of Epidemiology, Cardiovascular Disease Prevention and Health Promotion, National Institute of Cardiology, 04-628 Warsaw, Poland; wdrygas@ikard.pl; 6Department of Social and Preventive Medicine, Medical University of Lodz, 90-419 Lodz, Poland

**Keywords:** cardiac electrotherapy devices, hospital readmissions, outpatient care

## Abstract

*Background and Objectives*: Cardiovascular implantable electronic device (CIED) treatment is widely used in modern cardiology. Indications for this type of treatment are increasing. However, a significant proportion of CIED implantation patients require subsequent hospitalization for cardiovascular reasons. Older age and the associated complex clinical picture necessitate multidisciplinary outpatient specialist care for these patients. The aim of this study was to analyze the reasons for subsequent hospitalizations in the cardiology department and the impact of outpatient specialty care on these hospitalizations. To the best of our knowledge, there are no such studies in the available literature. *Materials and Methods*: This study was conducted on a population of patients treated with CIED. Reasons for subsequent hospitalizations were divided into clinically and statistically valid groups according to the main diagnosis. Using an electronic database, causes of hospitalization were determined based on this diagnosis. Using data on consultations at outpatient specialty clinics, a logistic regression model was created for the probability of subsequent hospitalization for cardiovascular causes according to the specialty of the clinic. *Results:* The 9-year follow-up included a population of 2071 patients treated with CIED. During the follow-up period, 508 patients (approximately 24.5%) required subsequent hospitalization for cardiovascular reasons. The most common leading causes were heart failure, atrial fibrillation, and coronary artery disease. The need for consultation at outpatient specialty clinics increased the likelihood of hospitalization. Moreover, the need to consult patients in nephrology outpatient, pulmonary disease outpatient, and orthopedic outpatient clinics was the most significant. *Conclusions:* The use of electronic implantable cardiovascular devices is a very important part of therapy in modern cardiology. The methods for their use are constantly being improved. However, they represent only one stage of cardiac treatment. After CIED procedures, patients require further care in both inpatient and outpatient specialty care settings. In this paper, we outline the reasons for subsequent hospitalizations and the importance of outpatient specialty care in this context. Effective organization of care after CIED procedures may be important in reducing the most expensive component of this care, that is, inpatient treatment.

## 1. Introduction

Treatment with a cardiovascular implantable electronic device (CIED) is a very valuable method of management in modern cardiology. Indications for this therapy are constantly expanding. The latest guidelines from the European Society of Cardiology (ESC) concerning cardiac pacing and cardiac resynchronization therapy (CRT) were published in 2021 [1]. The indications for the use of implantable cardioverter defibrillators (ICDs) in heart failure were presented in the latest ESC guidelines, published in August of 2021 [2]. Data on the use of ICD in the prevention of sudden cardiac death and ventricular arrhythmias are somewhat older [3,4]. The population of patients treated with these methods is becoming older and thus more burdened with comorbidities. Despite modern treatment methods, the prognosis of these patients remains serious. This is especially true for patients with implanted single-chamber pacing systems and after generator replacement, as we demonstrated in an earlier paper [5]. The implantation of the device also has a very important diagnostic function. Modern systems allow for the diagnosis of subclinical atrial fibrillation. One review found that approximately one-third of patients with risk factors for stroke but without atrial fibrillation at the time of implantation were diagnosed with paroxysmal subclinical atrial fibrillation within 3 years. This is of considerable clinical significance [6]. System implantation is only one step in the treatment of these patients. The complexity of the clinical picture is one of the reasons for subsequent hospitalizations for cardiac reasons. These patients also require specialist outpatient treatment in various outpatient clinics. The purpose of this study was to describe the reasons for subsequent cardiac hospitalizations in this population and the impact of the need for outpatient specialty care on these hospitalizations.

## 2. Materials and Methods

### 2.1. Study Population

The present single-center analysis was carried out based on data acquired from the Electrophysiology Laboratory of the Masovian Specialist Hospital in Ostroleka, where approximately 300 procedures of this type are performed annually. Over the course of nine years (2010–2018), the analysis considered a large group of patients with a history of implantation of single- and double-chamber pacemaker devices, cardioverter defibrillators, and cardiac resynchronization devices.

Using an electronic database and hospital treatment information sheets, data were collected on subsequent hospitalizations after discharge following device implantation. The data on reasons for rehospitalization based on main diagnosis come from the database we created for the study population.

Hospital treatment information sheets included the main reason for hospitalization and comorbid diagnoses. Reasons for subsequent hospitalizations were divided into clinically and statistically valid groups according to the main diagnosis (Group 1: heart failure (HF); Group 2: atrial fibrillation (AF); Group 3: coronary artery disease (CAD); Group 4: other cardiac arrhythmias; Group 5: infective endocarditis; Group 6: other causes). A multivariate logistic regression model including data on outpatient specialist consultations, certain clinical data, age, and length of hospitalization associated with device implantation was used to estimate the chances of another cardiac hospitalization.

### 2.2. Statistical Analysis

In the statistical analysis, the chi-square independence test was used to check the relationship between quality characteristics. The normality of distribution was verified using Kolmogorov–Smirnov tests, with Lillefors correction and the Shapiro–Wilk test. A nonparametric Kruskal–Wallis one-way ANOVA on ranks test with a post-hoc test of multiple comparisons of mean ranges for all samples was used to compare quantitative variables without normality of distribution in the case of many groups. Survival curves were estimated using the Kaplan–Meier method. Differences between survival curves were assessed using the log-rank test. Statistically significant results were considered at *p* < 0.05. In order to carry out the calculations, Statistica 13.3 (Statsoft Inc. Tulsa, OK, USA) and Stata/IC 13 (StataCorp, College Station, TX, USA) were used.

## 3. Results

A population of 2071 patients undergoing cardiac electrotherapy device implantation was analyzed regarding rehospitalization for cardiac reasons. A total of 508 patients required at least one hospitalization. This population was slightly younger and predominantly male. Patients with implanted ICD/CRT systems required rehospitalization more frequently. Among the comorbidities, atrial fibrillation, chronic obstructive pulmonary disease (COPD), and coronary artery disease were significantly more frequent. The characteristics of the group requiring hospitalization as compared with those not requiring further hospital treatment at the cardiology department are shown in Table 1. Table 2 shows the primary indications for device implantation.

The long-term survival of rehospitalized and non-rehospitalized patients (days; median) was not significantly different (2966 vs. 2965). The survival curves of patients who were rehospitalized and those not requiring further hospitalization after device implantation are shown in Figure 1.

The main diagnoses during readmissions in the study group (*n* = 508) were as follows: HF (*n* = 301; 59%); AF (*n* = 87; 17%); CAD (*n* = 43; 9%); other arrhythmias (*n* = 37; 7%); infective endocarditis (*n* = 9; 2%); other causes (*n* = 31; 6%);

Survival of patients depending on the main cause of subsequent hospitalization is shown in Figure 2.

During the study period, patients requiring rehospitalization had 408 consultations at a cardiology outpatient clinic, 91 at a urology outpatient clinic, 89 at an orthopedic outpatient clinic, 80 at an ophthalmologic outpatient clinic, 79 at a vascular surgery outpatient clinic, 74 at a general surgery outpatient clinic, 61 at a lung disease outpatient clinic, and 57 at a nephrology outpatient clinic.

A multivariate logistic regression model assessing the chance of subsequent rehospitalization after CIED implantation (taking into account the need for a specialist outpatient clinic consultation, certain clinical data, age at implantation, and length of implant-related hospitalization) is presented in Table 3. On the basis of the analysis performed, it was found that diseases requiring specialist nephrological, pulmonological, and orthopedic treatment had the greatest impact on the necessity of readmission to hospital for the above reasons.

## 4. Discussion

### 4.1. Most Common Causes of Rehospitalizations for Cardiovascular Causes

The problem of rehospitalizations affects all healthcare systems, and is an important factor in increasing the overall cost of treatment. The excessive number of hospitalizations and the focus on expensive hospital care upsets the balance between the inpatient and outpatient sectors. Cardiovascular diseases are the most common causes of hospitalization in Poland, at a rate of 13%. Men are twice as likely to be hospitalized for diseases related to atherosclerosis, and only slightly less frequently for cardiovascular diseases, chronic kidney disease, liver diseases, and injuries. The share of cardiovascular diseases increases as the population ages. These ratios have increased by 25% in recent years. The incidence of hospitalization in Poland is similar to that in other EU countries. In recent years, there has been an increase of 30% [7].

In this study, we analyzed the cardiovascular causes of the rehospitalization of patients after CIED treatment. In the medical literature, there is a shortage of studies on the problem of rehospitalization for cardiac reasons in patients with implanted devices for heart electrotherapy. The available studies include only short- and mid-term analyses of rehospitalizations following high-energy device (ICD, CRTD) implantations. As in the present paper, the most common cause of hospitalization was heart failure [8]. Patients with ischemic heart disease were at higher risk of further hospitalization [9].

Hospitalizations due to HF account for 1–2% of all hospital admissions [10]. Approximately half of patients are admitted at least once a year after diagnosis, and this figure increases to approximately 80% within 5 years [11,12]. As a result of the ageing of the population, this percentage is projected to increase steadily over the coming years. A six-month analysis of the reasons for rehospitalization of patients with heart failure by Umehara et al. showed that CKD, dementia, and low motor performance were risk factors for rehospitalization [13].

Heart failure represents a new 21st century epidemic. The most common cause is ischemic heart disease (70%). The number of patients with heart failure in Poland exceeds 750,000, and forecasts predict an increase of about 25% within a decade [14].

This disease is the most common cause of hospitalization in patients above 65 years of age in Poland. The median age of the study population was 76 years. For this reason, hospitalization rates are among the highest in Europe (twice as those in OECD countries; five times higher than those in the UK) [15].

The most common cause of rehospitalization was heart failure; however, in Table 3, it is shown that COPD had a higher OR than HF in causing rehospitalization. Table 3 includes some of the clinical data including comorbidities with which patients in the study population were burdened during hospitalization related to implantation of a cardiac electrotherapy device. Reasons for rehospitalization were determined based on the principal diagnosis (data were derived from hospital treatment information sheets). Being under the care of an outpatient lung disease clinic significantly increased the chance of rehospitalization for cardiac reasons. Among these patients, many were burdened with COPD. Hospitalization costs account for more than 90% of the system’s expenditure on the treatment of heart failure. Five-year survival is comparable or worse than that for most cancers. The coordination of post-hospital care may involve a reduction in the number of heart failure hospitalizations by approximately 34%, and the total number of hospitalizations by approximately 27% [16].

In the present 9-year analysis, more than 38% of patients experienced atrial fibrillation. AF was the second most common cause of rehospitalization after this kind of treatment. The available literature lacks data on the prevalence of AF in the population with implanted electrotherapy devices and the need for rehospitalization for this reason.

Medical progress has improved the quality of life of these patients, but there is an observable increase in the frequency of hospitalizations for this reason. This generates increasing treatment costs [17,18]. A total of 15–18% of patients require rehospitalization within 30 days of discharge [17,19]. Atrial fibrillation is one of the most common arrhythmias. As a result of the ageing of the population, its frequency is increasing. The prevalence of this disease is underestimated due to its mild or asymptomatic course in many patients. This arrhythmia is of very high public health significance because of the significant impact on mortality due to ischemic stroke, heart failure, and acute coronary syndromes [6,20].

The most common causes of readmissions presented in the present study are pathophysiologically related. Atrial fibrillation, coexisting with hypertension, metabolic syndrome, dyslipidemia, and diabetes, accelerates the development of atherosclerosis. Heart attack is a fairly typical element of the natural history of atrial fibrillation. Ischemic injury is the most common cause of heart failure. A large proportion of patients with heart failure suffer from atrial fibrillation.

### 4.2. Rehospitalizations for Cardiovascular Causes—The Impact of Outpatient Specialty Care

The present study also analyzed the impact of specialist outpatient care on hospitalizations for cardiac reasons. Studies of this type have not been published until now. The population of 508 patients was consulted in various specialist outpatient clinics during the follow-up period. In total, 939 consultations were held in the clinics of our hospital. We showed that patients under the care of certain specialty clinics were more likely to have subsequent hospitalizations for cardiovascular causes. This may be due to the fact that comorbidities that increase the chances of hospitalization are within the focus of these outpatient clinics, but may also reflect inefficient organization of the health care system. In rural areas, access to specialty care is worse than in urban areas. The need for hospitalization may have resulted from a lack of coordination between different sectors of medical care. The study group had a predominantly elderly population. These patients were burdened with various comorbidities. A total of 15% of those rehospitalized for cardiovascular reasons had chronic kidney disease. CKD affects approximately 10% of the adult population and is associated with increased morbidity and mortality. There were also high rates of hospitalization in this group [21,22,23,24]. One in four patients with CKD was hospitalized due to heart failure [25]. In our study, we showed that CKD, as the most common reason for needing care in an outpatient nephrology clinic, had an impact on hospitalizations for cardiovascular causes. This result is caused by the fact that our analysis was retrospective and confirms influence of concomitant diseases on patient prognosis including hospitalizations. CKD remains, also in populations of patients with cardiac implantable devices, one of the strongest factors negatively influencing health status independently. According to the created model, an increased chance of hospitalization for cardiological reasons was shown for several clinics (nephrology, lung diseases, orthopedic, vascular surgery, general surgery, ophthalmology, urology, and cardiology). The greatest chance of hospitalization occurred in the population of patients consulted in the nephrological dispensary and lung disease outpatient clinic.

Older age, multiple morbidity, and a lack of coordination of specialist care may explain the significant increase in the chance of further hospitalization.

In recent years, the lack of coordination in ambulatory specialist care and the associated negative consequences in the form of poor clinical outcomes have come to light. This also causes low patient acceptance [26,27,28,29,30,31]. The coexistence of comorbidities, typical of ageing societies, exacerbates the effects of the lack of coordination of care and increases the incidence of hospitalization. In the older population, multiple morbidity is common—it is estimated at 50–85% [32,33].

While ambulatory specialist care is not optimally coordinated, there is also a lack of proper communication between outpatient specialist care and primary care. This can be attributed to various factors, including a lack of regulation, as well as a lack of integration of information systems in these healthcare sectors. In a study by Kailasam et al., the greatest fragmentation was found in hematology, endocrinology, and anesthesia clinics [34]. The lack of coordination of specialist care may lead to further hospitalizations.

### 4.3. Rehospitalizations for Cardiovascular Causes—The Importance of Some Comorbidities

Among comorbidities, COPD and the HF III NYHA class were shown to be the most significant in increasing the odds of hospitalization for cardiovascular reasons. In the present study, the rate of heart failure among patients with COPD was 70.5%, and the rate of COPD in patients with HF was 17.6%. This was less common in patients who did not require further hospitalization. Patients controlled at the Lung Disease Clinic were more than twice as likely to be hospitalized for cardiac reasons. Among patients requiring rehospitalization, the proportion of patients burdened with COPD was higher than the average for Poland and amounted to 12% (not requiring hospitalization: 6.4%). The incidence of heart failure in patients with COPD ranges from 10% to 46% [35], and the incidence of COPD in patients with HF is approximately 10–20%. This proportion increases in older age groups [36,37]. The greatest risk of rehospitalization is associated with the coexistence of chronic obstructive pulmonary disease with chronic heart failure and osteoporosis [38]. COPD is one of the most common chronic diseases (7.6%), and represents an increasing economic burden on healthcare systems [39,40]. In the Polish population, the prevalence of COPD among patients above 40 years of age is larger, and amounts to about 10%. The risk of developing heart failure among COPD patients is almost five times higher than that in the general population [41]. In the work Kaszuba et al., it was shown that if HF and COPD occur at the same time, other cardiovascular diseases aggravating the prognosis of these patients are more often observed [42]. During an exacerbation of COPD, new or worsening cardiac arrhythmias (e.g., atrial fibrillation) occur. This is sometimes an indication for hospital treatment for cardiological reasons. As the respiratory function of the lungs deteriorates due to the development of pulmonary hypertension, right ventricular heart failure develops.

### 4.4. Survival of Patients According to Rehospitalization and Those Not Requiring Hospitalization

The statistically insignificant lack of difference in survival between patients requiring rehospitalization for cardiac reasons and those not requiring rehospitalization, obtained in the present study, may have been due to the specificity of the health care system in an area with predominantly rural areas. Due to difficult access to outpatient specialist care, the cardiology department had to take over some of the functions of outpatient specialist care. Patients were admitted to hospital more often but in less advanced stages of the disease. This could have a positive prognostic significance. In an analysis evaluating access to medical services in rural areas, it was found that hospitalization rates increase according to the rurality of the area in which patients reside. The average length of hospitalization decreases. A higher percentage of patients receives care in hospitals, among others, in Hospital Emergency Departments and primary care clinics. This adversely affects the health status of this population [43]. In the available literature, the prognosis of patients requiring hospitalization is worse regardless of the severity of heart failure [12].

### 4.5. Survival of Patients Depending on the Main Cause of Subsequent Hospitalization

We analyzed the survival of patients according to the criteria used in the analysis of the frequency of rehospitalizations. As expected, the group with heart failure as the main diagnosis was burdened with the worst prognosis, later, in contrast to that analysis, were coronary artery disease and atrial fibrillation as main diagnoses.

There are few long-term studies of this type in the available literature. The 3-year study by Lamblin et al. was conducted on a younger population than in our analysis. A total of 70.6% of patients had only one disease burden at the time of inclusion in the analysis. Mortality was highest for HF (27%). Mortality rates for atrial fibrillation and coronary artery disease were 17.5% and 12.2%, respectively [44].

Almost half of the patients in the group analyzed were burdened with heart failure already before implantation of CIED and over 41% with coronary artery disease, and atrial fibrillation was present in almost 40% of patients. Other relevant factors were hypertension (⅔ of patients) and diabetes (almost ⅓ of patients). Prevalence of these concomitant diseases is higher than in the general population.

The prevalence of heart failure is increasing in ageing populations [45,46]. Over the age of 70, the prevalence is about 10% [47,48]. The prognosis is still poor despite continuous improvement associated with the introduction of modern methods of treatment according to constantly updated guidelines. The prognosis is worse in observational studies than in clinical trials. A study combining the Framingham Heart Study and Cardiovascular Health Study Cohort reported a 67% mortality rate within 5 years of diagnosis [49].

Coronary artery disease is one of the leading reasons for the prevalence and mortality in the older population. There are few papers in the available literature that analyze the diagnosis, management, and course of CAD in older age groups. Older patients are often excluded from clinical trials [50]. CAD’s dissemination grows with the age of patients, constituting 10–12% in women aged 65–84 and 12–14% in men of the same age. The yearly mortality depends on the complexity of the clinical picture. In case of numerous burdens, peripheral arterial disease, history of heart failure and diabetes, it is about 3.8% [51].

Like in the case of HF or CAD, the incidence of atrial fibrillation is constantly increasing. It is estimated that the incidence of atrial fibrillation in adults reaches between 2 and 4 per cent, and an increase by 2.3 times is expected [52].

Age is an independent risk factor for the occurrence of this arrhythmia. In 2010, the population with atrial fibrillation was estimated at 5.6 million in Western Europe. Based on the analyses, it is estimated that in 2060 this population may reach 13.8 million people. Patients over 80 years of age will constitute the majority of this group (65.2%) [53,54].

The connection of atrial fibrillation and increased mortality is well documented. All-cause mortality risk is two times higher in women and 1.5 higher in men with atrial fibrillation [55,56,57]. Similarly, the increased mortality risk is observed in patients with other cardiovascular system diseases. Atrial fibrillation significantly increases the risk of sudden cardiac death both in the general population and in patients with coronary artery disease, heart failure and patients with CIED [58].

### 4.6. Limitations

The present research has its limitations, including its single-center nature and the lack of a broader clinical profile for all patients (e.g., pharmacotherapy data). Another limitation of the present work is that laboratory parameters were not included, which might, to some extent, allow for predictions of the outcome. Nonetheless, all patients were assessed according to nonheart-related comorbidities, which form a certain equivalent to positive or negative laboratory results. Another limitation of current analysis is insufficient data on concomitant chronic illness, especially in frail elderly and cardiological patients that frequently require hospitalization in real life. This issue is currently under consideration, especially in chronic heart failure and the COVID-19 era. This aspect needs further investigation analysis.

## 5. Conclusions

Despite the application of modern treatment methods, such as the use of CIED, patients required further specialist care after the performed procedures. A significant number had to be re-admitted to hospital for cardiological reasons (most frequently heart failure, atrial fibrillation, and coronary artery disease). The complexity of the clinical picture was also the reason for the need for consultations in outpatient specialist clinics. On the basis of the analysis performed, it was found that diseases requiring specialist nephrological, pulmonological, and orthopedic treatment had the greatest impact on the need for rehospitalization for the above reasons. Effective organization of care after CIED procedures may be important in reducing the most expensive component of this care, that is, inpatient treatment.

## Figures and Tables

**Figure 1 medicina-58-00151-f001:**
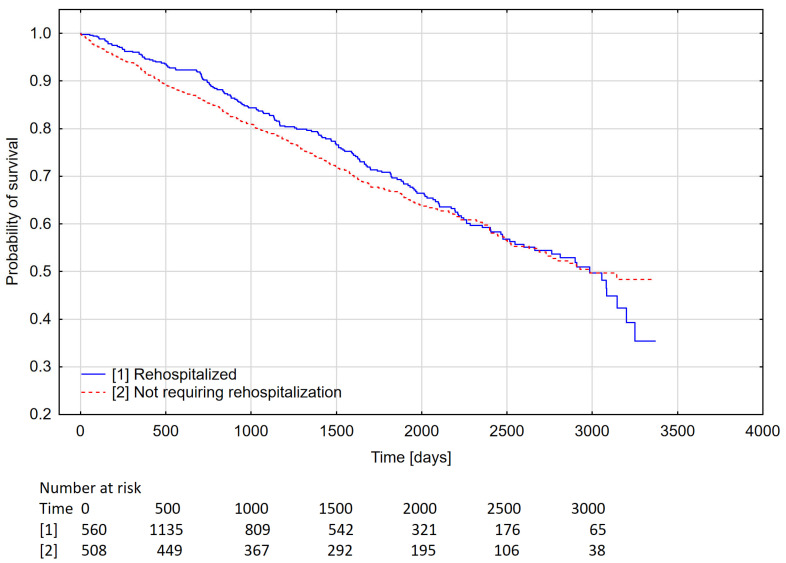
Kaplan–Meier curves showing the survival of rehospitalized and non-requiring rehospitalization patients following cardiovascular implantable electronic device implantation (*p* = 0.392).

**Figure 2 medicina-58-00151-f002:**
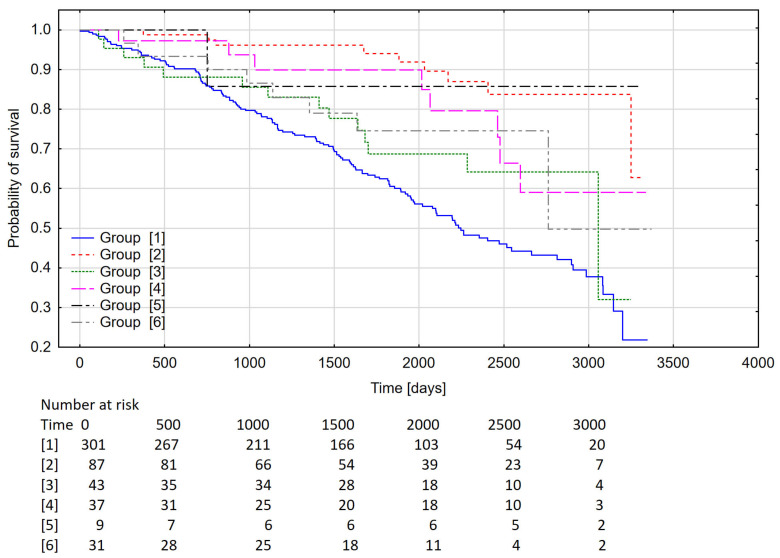
Kaplan–Meier curve representing the probability of survival for the groups of diagnoses (*p* < 0.001). Group 1: heart failure; Group 2: atrial fibrillation; Group 3: coronary artery disease; Group 4: other cardiac arrhythmias; Group 5: infective endocarditis; Group 6: other cases.

**Table 1 medicina-58-00151-t001:** The characteristics of groups of patients rehospitalized versus those not requiring rehospitalization following CIED implantation (data presented as numbers and percentages).

Variable	Rehospitalized Patients *n* = 508	Patients Not Requiring Rehospitalization *n* = 1563	*p*-Value
Age (years)	76.0 (68.0–82.0)	78.0 (70.0–83.0)	<0.001 *
Male	283 (55.7%)	794 (50.1%)	0.054
SC-AAI/VVI	160 (31.5%)	530 (33.9%)	<0.001 *
DC-DDD	251 (49.4%)	856 (54.8%)	
ICD/CRT	97 (19.1%)	177 (11.3%)	
Heart failure II NYHA class	146 (28.7%)	412 (26.4%)	0.293
Heart failure III NYHA class	98 (19.3%)	178 (11.4%)	<0.001 *
Hypertension	325 (64.0%)	1119 (71.6%)	0.001 *
Diabetes	155 (30.5%)	440 (28.2%)	0.307
Coronary artery disease	209 (41.1%)	579 (37.0%)	0.098
Dilated cardiomyopathy	27 (5.3%)	40 (2.6%)	0.002 *
Hypertrophic cardiomyopathy	2 (0.4%)	3 (0.2%)	-
Atrial fibrillation	177 (38.4%)	393 (25.1%)	<0.001 *
History of stroke	55 (10.8%)	160 (10.2%)	0.705
Chronic obstructive pulmonary disease (COPD)	61 (12.0%)	100 (6.4%)	<0.001 *
Chronic kidney disease (CKD)	76 (15.0%)	249 (15.9%)	0.601
Hyperthyroidism	22 (4.3%)	55 (3.5%)	0.401
Hypothyroidism	26 (5.1%)	78 (5.0%)	0.909
LVEF—primary prevention	29.0% (25.0–33.0)	30% (25.0–33.0)	0.652
LVEF—secondary prevention	37.0% (30.0–45.0)	39.0% (30.0–45.0)	0.570
Type of procedure	*n*	*n*	
First time implantation	471 (92.7%)	1461 (93.5%)	0.553

* The differences are statistically significant at *p* < 0.05. CIED, cardiovascular implantable electronic device; SC, single chamber; DC, dual chamber; AAI, atrial single-chamber pacemaker; VVI, ventricular single-chamber pacemaker; DDD, dual-chamber pacemaker; ICD, implantable cardioverter defibrillator; CRT, cardiac resynchronization therapy; NYHA, New York Heart Association; LVEF, left ventricular ejection fraction; AV, atrioventricular.

**Table 2 medicina-58-00151-t002:** Primary indications for device implantation (data presented as numbers and percentages).

Primary Indications	Rehospitalized Patients *n* = 508	Patients Not Requiring Rehospitalization *n* = 1563	*p*-Value
Atrial fibrillation with AV block	141 (27.8%)	393 (25.1%)	0.242
AV block III	54 (10.6%)	349 (22.3%)	<0.001 *
Sick sinus syndrome	151 (29.7%)	434 (27.8%)	0.395
AV block II t.2	43 (8.5%)	110 (7.0%)	0.285
AV block 2:1	18 (3.5%)	94 (6.0%)	0.032 *
Trifascicular block	1 (0.2%)	1 (0.06%)	-
AV block II t.1	8 (1.6%)	12 (0.8%)	0.106
Alternating bundle branch block	-	2 (0.1%)	-
Cardiac arrest—primary prevention	74 (14.6%)	131 (8.4%)	<0.001 *
Cardiac arrest—secondary prevention	23 (4.5%)	46 (2.9%)	0.084

* The differences are statistically significant at *p* < 0.05. AV, atrioventricular.

**Table 3 medicina-58-00151-t003:** Multivariate logistic regression model estimating the chances of rehospitalization following CIED implantation.

Consultations in Specialist Clinics	OR	95% CI	*p*-Value
Nephrology outpatient clinic	3.74	2.35–5.95	<0.001 *
Lung disease outpatient clinic	2.17	1.42–3.31	<0.001 *
Orthopedic outpatient clinic	2.02	1.42–2.87	<0.001 *
Vascular surgery outpatient clinic	1.77	1.23–2.57	0.002 *
General surgery outpatient clinic	1.72	1.18–2.50	0.005 *
Ophthalmological outpatient clinic	1.57	1.09–2.26	0.015 *
Urology outpatient clinic	1.43	1.02–2.02	0.039 *
Cardiology outpatient clinic	1.30	1.01–1.68	0.043 *
Clinical data			
Chronic obstructive pulmonary disease	1.76	1.25–2.50	0.001 *
Heart failure III NYHA class	1.59	1.20–2.12	0.001 *
AV block 2:1	0.55	0.33–0.93	0.024 *
AV block III	0.43	0.31–0.59	<0.001 *
Other factors			
Length of hospitalization	1.04	1.03–1.06	<0.001 *
Age	0.99	0.98–1.00	0.012 *

* The differences are statistically significant at *p* < 0.05. OR, odds ratio; CI, confidence interval; NYHA, New York Heart Association; AV, atrioventricular.

## Data Availability

The data presented in this study are available upon reasonable request from the corresponding author.

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
