# Peer review of "Long-Term Outpatient Care and Rehospitalizations in Patients after Cardiac Electrotherapy Device Implantation"

_medicina, 2022, doi:10.3390/medicina58020151_

Round 1
Reviewer 1 Report
In this study, Dr. Zaluska and collegues (if I am correct) tried to investigate the reasons for re-hospitalisation in the electrotherapy population. To be honest, this is not an easy manuscript to write. Almost everything is unclear and therefore require clarifications.
- The definition of electrotherapy is missing. I am not sure what is the idea of this paper. Is this the "electrotherapy" meant (https://en.wikipedia.org/wiki/Electrotherapy)?
- If so, why does it need to be implanted? I have never seen an implantation of a physiotherapy device before?
- The abstract and introduction do not help to understand what the authors tried to do here. Please rewrite everything and check other publications on how to write good abstract and introduction.
- Line 48: "However, the available literature lacks up-to-date data on re-hospitalisations and the burden of outpatient specialist advice." what is actually the aim of this study? I thought it is about the reason for hospitalizations? Please clarify.
- Line 30: "the most common underlying cause was..." needs to be followed by one cause only, not 3. Please change to "the most common underlying causes were..."
- "Hospitalizations were classified into six types depending on the underlying cause (Group 1 – heart failure (HF), Group 2 – atrial fibrillation (AF), Group 3 – coronary artery disease (CAD), Group 4 – other cardiac arrhythmias, group 5 – other causes, Group 6 – infectious endocarditis)." What is the point of having these groups? They are only used once in the Kaplan-Meier, but the absence of these groups, there is no effect, really.
In general, at this point, it is very difficult to provide specific comments and suggestions for this manuscript because the background, the target population, the research aim/question, the variable of interest, and many more are unclear. I would generally suggest to rewrite the abstract, introduction and methods parts before submitting this again. Please check the instruction for authors for a guidance in writing those parts or consult with someone who has a writing experience (https://www.mdpi.com/journal/medicina/instructions)
Also, please make sure that the template is correct. Submitting a manuscript with IJERPH template to Medicina somewhat reflects the low efforts put into the writing.
Author Response
Dear Reviewer,
Thank you very much for your valuable comments and suggestions on how to improve the readiness and overall quality of our manuscript. We also thank you for giving us the opportunity to rewrite some parts of the manuscript and clarify any doubts. We believe that the problem presented in the paper is original and important from both cardiology and public health perspectives.
We used MDPI to improve the language side of our manuscript. We ordered a specialist edit of our text. This service was confirmed by a certificate.
Change tracking was used when applying the corrections. The corrected text was written in red font.
In this study, Dr. Zaluska and collegues (if I am correct) tried to investigate the reasons for re-hospitalisation in the electrotherapy population. To be honest, this is not an easy manuscript to write. Almost everything is unclear and therefore require clarifications.
- The definition of electrotherapy is missing. I am not sure what is the idea of this paper. Is this the "electrotherapy" meant (https://en.wikipedia.org/wiki/Electrotherapy)?
If so, why does it need to be implanted? I have never seen an implantation of a physiotherapy device before?
Thank you for your comment. We agree that the term used is not clear. We will introduce the term Cardiovascular Implantable Electronic Device (CIED) instead of electrotherapy in the text. The article describes the most common implantable devices used in cardiology according to current guidelines (single- and dual-chamber pacemakers, implantable cardioverter-defibrillators, and cardiac resychronization devices).
- The abstract and introduction do not help to understand what the authors tried to do here. Please rewrite everything and check other publications on how to write good abstract and introduction.
Thank you for this valuable comment. We agree that the indicated sections should be revised to improve the readability of the paper. The abstract, introduction, and methods have been rewritten as recommended.
- Line 48: "However, the available literature lacks up-to-date data on re-hospitalisations and the burden of outpatient specialist advice." what is actually the aim of this study? I thought it is about the reason for hospitalizations? Please clarify.
Thank you for this comment. The purpose of this study was a long-term analysis of hospital readmissions for cardiovascular causes that occurred in a population of patients with implantable CIED devices. It also analyzed the effect of outpatient specialty care on increasing the likelihood of hospitalization for cardiovascular causes.
In accordance with this valuable suggestion, the purpose of the study was further clarified in the Abstract and Introduction.
This topic was taken up by the authors due to the lack of such studies in the available literature.
- Line 30: "the most common underlying cause was..." needs to be followed by one cause only, not 3. Please change to "the most common underlying causes were..."
Thank you for this comment. Text corrected.
- "Hospitalizations were classified into six types depending on the underlying cause (Group 1 – heart failure (HF), Group 2 – atrial fibrillation (AF), Group 3 – coronary artery disease (CAD), Group 4 – other cardiac arrhythmias, group 5 – other causes, Group 6 – infectious endocarditis)." What is the point of having these groups? They are only used once in the Kaplan-Meier, but the absence of these groups, there is no effect, really.
The hospital treatment information sheets included the main cause of hospitalization and comorbid causes. The division into groups with the main focus on the principal diagnosis seemed clinically justified and allowed statistical analysis.

Reviewer 2 Report
authors report a very interesting report on patients that need re-hospitalization after cardaic electrotherapy.
the article is well written and results are clearly exposed.
in the background authors well describe that the population of western countries is ageing requiring in older age frequent hospitalization.
Yet, the clinical concept of frail elderly or of frail cardiological patients is missing in their results as far as the number of patients with 3 or more chronic illness that frequently require hospitalization in the real life
Author Response
Dear, Reviewer
Yet, the clinical concept of frail elderly or of frail cardiological patients is missing in their results as far as the number of patients with 3 or more chronic illness that frequently require hospitalization in the real life
Thank you very much for your positive assessment of our article. The remark is very right. This topic is being considered for another paper.

Round 2
Reviewer 1 Report
Thank you for providing responses to my previous queries. Overall, although the manuscript is slightly better, there are some issues that need to be addressed by the authors:
- Line 118: "infectious endocarditis" should be "infective endocarditis"
- Please always add the total number of patients included in each group (rehospitalized vs. non-rehospitalized) in Table 1 and 2. Otherwise, the percentages provided in the table are meaningless and it is hard to interpret the tables.
- In all Kaplan-Meier, please make sure to add the number of patients included and also the number at risk in every time points.
- Also, please specify the unit of the x-axis of the Kaplan-Meier plots. It does not make sense to have 3500 years of observation. Are they in months?
- "HF (59%);, AF (17%);, CAD (9%);, other arrhythmias (7%);, other causes (6%);, infectious endocarditis (2%)." How did the authors get these numbers? Which tables represent these values?
- It is weird to see that half of the discussion relies on the data presented above, while the rest of the findings are not adequately discussed. For example, Table 2 was not even discussed in the discussion. So, what is the point of having this table?
- I would suggest to rewrite / reorganize the whole discussion section to better explain the findings of this manuscript. Please write a constructive, focused and valid discussion as this is one of the most important parts of this manuscript.
- So far, the authors discussed and made connections with many things that are not shown in the paper, rationalizing their data to fit those arguments. This is kind of inappropriate.
- I am not sure what is the importance of having "Consultations in specialist clinics" in table 3? What is the insight we can get from it?
- In the discussion, the authors said that "the most common cause of hospitalization was heart failure", however, in Table 3, it is shown that COPD had a higher OR than HF in causing rehospitalization. Please discuss this data discrepancy in details. What would be the possible cause(s) of this inconsistency?
- Line 243: Remove "in"
- "The presented study also analyzeds the impact of specialist outpatient care on the frequency of hospitalizations for cardiac reasons." Which part of the findings reported this? I could not find anything about frequency of hospitalization in this manuscript.
- "In this present study, the older population was found to dominates, especially in the group with implanted cardiac pacemakers." which data supported this statement? There is no data on the non implanted group in this manuscript.
- The absence of difference of survival between rehospitalized vs. non-rehospitalized patients observed in kaplan-meier plots (Figure 1) should also be extensively discussed.
- Similarly, the survival of those groups in Figure 2 has to be discussed. Why the survival of AF is better than CAD, for instance?
- "In the presented analysis, 15% of patients requiring re-hospitalization were suffering from CKD, and control at a nephrology outpatient clinic was the factor with the greatest impact on subsequent hospitalizations." So, what does it mean? The nephrologists were not good enough to control the CKD and therefore, rehospitalization was high in patients who visited nephrology clinic?
- Please provide the clean / non-tracked version in the next iteration for a thorough check. It is too distracting to have all of those minor language corrections.
- Also, please recheck for typos and minor language errors that are still present throughout the manuscript.
Reviewer 2 Report
the answer to my question had a positive from authors although they did not replied in the text because the topic is ongoin in another manuscript.
I could be in agreement with this but in truth they should mention this missing point as a study limitation to improve the manuscript. 2 or sentences in the discussion may be sufficient.
